# Determinants of 1-Year Adverse Event Requiring Re-Hospitalization in COVID-19 Oldest Old Survivors

**DOI:** 10.3390/geriatrics8010010

**Published:** 2023-01-10

**Authors:** Chukwuma Okoye, Riccardo Franchi, Alessia Maria Calabrese, Virginia Morelli, Umberto Peta, Tessa Mazzarone, Igino Maria Pompilii, Giulia Coppini, Sara Rogani, Valeria Calsolaro, Fabio Monzani

**Affiliations:** 1Geriatrics Unit, Department of Clinical & Experimental Medicine, University Hospital of Pisa, 56121 Pisa, Italy; 2Aging Research Center, Department of Neurobiology, Care Sciences and Society, Karolinska Institutet and Stockholm University, SE-106 91 Stockholm, Sweden

**Keywords:** COVID-19 survivors, elderly, frailty, rehospitalization, long COVID

## Abstract

The incidence of “Long COVID” syndrome appears to be increasing, particularly in the geriatric population. At present, there are few data regarding the relationship between long COVID and the risk of re-hospitalization in the oldest old survivors. Patients older than 80 years consecutively hospitalized for COVID-19 in our tertiary care hospital were enrolled and followed after discharge in a 12-month ambulatory program. A comprehensive geriatric assessment (CGA), including functional capabilities and physical and cognitive performances, was performed at 6-month follow-up. Frailty degree was assessed using a 30-item frailty index. The re-hospitalization rate was assessed at 12-month follow-up through a computerized archive and phone interviews. Out of 100 patients discharged after hospitalization for COVID-19 (mean [SD] age 85 [4.0] years), 24 reported serious adverse events requiring re-hospitalization within 12 months. The most frequent causes of re-hospitalization were acute heart failure (HF), pneumonia and bone fracture (15.3% each). By multivariate logistic analysis, after adjustment for potential confounders, history of chronic HF [aOR: 3.00 (CI 95%: 1.10–8.16), *p* = 0.031] or chronic renal failure [aOR: 3.83 (CI 95%: 1.09–13.43), *p* = 0.036], the burden of comorbidity [(CIRSc) aOR: 1.95 (CI 95%: 1.28–2.97), *p* = 0.002] and frailty [aOR: 7.77 (CI 95%: 2.13–28.27), *p* = 0.002] resulted as independent predictors of re-hospitalization. One-fourth of the oldest old patients previously hospitalized for COVID-19 suffered from adverse events requiring re-hospitalization, two-thirds of them within three months after discharge. Frailty, the burden of comorbidity, history of chronic HF or chronic renal failure, but not COVID-19 disease severity, independently predicted re-hospitalization.

## 1. Introduction

Recently, the study and understanding of short- and long-term health consequences of COVID-19 has assumed a role of primary importance. It is known that SARS-CoV-2 infection causes systemic disease with potential impairment of respiratory, digestive, cardiovascular, renal, immune and nervous systems [1].

A considerable number of patients complain of symptoms related to COVID-19 even weeks after the onset of the disease, a condition described as post-acute COVID-19 syndrome or “Long COVID” [2,3,4]. Long COVID incidence appears to be increasing particularly in the geriatric population, but, at present, there are little data regarding its relationship with the risk of re-hospitalization of the oldest old [5,6,7].

In older patients, long COVID can express itself through worsening cognitive status and functional decline, thus increasing the risk of re-hospitalization [8]. Of note, previously hospitalized COVID-19 survivors are at high risk for re-hospitalization with an estimated prevalence rate of 10% [9,10]; in particular, advanced age seems to be associated with post-discharge all-cause mortality even though conflicting data are still standing in the literature [9,10,11]. Interestingly, several studies reported an increased short-term excess mortality following SARS-CoV-2 infection, regardless of disease severity; however, scanty data are available regarding the main causes of re-hospitalization and death in oldest old patients.

Our study aims to: (i) evaluate the prevalence and causes of 12-month adverse events requiring re-hospitalization in a cohort of oldest old patients who recovered from SARS-CoV-2 infection and (ii) determine risk factors of adverse events requiring re-hospitalization. Furthermore, we evaluate the possible impact on functional and cognitive performance among re-hospitalized patients and controls.

## 2. Materials and Methods

In this single-center, prospective study, we evaluated a cohort of geriatric patients previously hospitalized for COVID-19 disease in a 12-month follow-up program designed to evaluate their overall recovery or persistence of long COVID-related symptoms. The study complied with the Declaration of Helsinki and was approved by the local institutional Ethics Committee. Each patient provided written informed consent to participate in the study. Data regarding the acute phase (the interval between symptoms onset to discharge from the hospital) were taken retrospectively from the hospital computerized archive and phone interviews. Demographic characteristics and clinical history were recorded. At six-month follow-up, the enrolled participants were evaluated by 2 trained physicians in an ambulatory setting. At one-year follow-up, mortality and re-hospitalization were recorded. A comprehensive geriatric assessment (CGA) [12] was performed, including cognitive evaluation using the Short Portable Mental Status Questionnaire (SPMSQ) [13], the autonomy level in terms of independence in the performance of basic (ADL) [14] and instrumental (IADL) [15] activities of daily living, and the risk of malnutrition using the Mini Nutritional Assessment-Short Form (MNA-SF) [16] and Body Mass Index (BMI). Comorbidity was assessed with the Cumulative Illness Rating Scale (CIRS-c) [17] while the degree of frailty was assessed using a 30-item frailty index (Appendix A). Persistence of symptoms was tested with 27 pre-identified symptoms on a checklist, with the patients reporting or not reporting their presence. In order to investigate muscle strength, the handgrip strength (HGS) test was performed using a hand dynamometer with the dominant hand. HGS is a simple measure of strength and may be utilized as a marker of sarcopenia. The cut-off points of <20 kg in women and <30 kg in men have been identified to detect patients at risk for sarcopenia. Participants were seated with shoulder adducted, elbow flexed to 90 degrees, and forearm and wrist neutral. The highest score of three consecutive measurements was recorded. The level of physical fitness was evaluated using the Short Physical Performance Battery (SPPB) [18], which measures physical function using 3 components: usual gait speed over 4 m, time to complete 5 chair rises and standing balance with progressively narrow base of support. Each component was scored on a 0–4 scale and summed for an overall range of 0–12.

Statistical analysis was performed with IBM SPSS Statistic (IBM SPSS Statistic v27.0 lnk, Armonk, NY, USA: IBM Corporation and its licensor 1989–2020). Continuous variables were presented as mean and standard deviation, ordinal variables as median and interquartile range (IQR), and categorical variables as percentage. The Mann–Whitney U test and chi-square test were used for multiple comparisons. Multivariate logistic regression analysis was performed to identify factors associated with adverse events.

A multivariate logistic regression was performed using a priori selected model covariates on the basis of clinical considerations. Covariates included were age, sex, BMI, smoking status, history of heart failure, chronic renal disease, hypertension, diabetes mellitus, prior stroke or transient ischemic attack and coronary artery disease. Probability for removal of variables in the model was set at *p* = 0.10 or higher. Estimated odds ratios (ORs) with 95% confidence intervals (CIs) were obtained. Tests were performed considering a level of significance of 5%.

## 3. Results

The baseline characteristics of the study population are shown in Table 1. Overall, 100 older patients [mean age 85 years (SD = 4), 42% women] previously hospitalized for COVID-19 were enrolled; of them, 24 reported serious adverse events requiring hospitalization. There were 6 patients re-hospitalized during the first month after discharge, 6 during the second, 4 during the third, 4 between the third and the sixth and 4 from the sixth until the twelfth month after discharge.

The most frequent causes of re-hospitalization were as follows: heart failure (15.3%), pneumonia (15.3%), bone fracture (15.3%), acute kidney failure (11.5%), peripheral neuropathy (11.5%) and sepsis (7.6%) (Table 2).

At baseline, patients who reported complications were more frequently men (58.3%) with a significant higher degree of disability compared to counterparts, both in basic [median BADL 5 (IQR = 3.25) vs. 6 (IQR = 1), respectively; *p* = 0.02] and instrumental [median IADL 4.5 (IQR = 6) vs. 7 (IQR = 4), respectively; *p* = 0.003] activities of daily living, along with a greater burden of comorbidities [median CIRS-c 4 (IQR = 1.25) vs. 2 (IQR = 2), respectively; *p* = 0.002] and a higher degree of frailty [mean frailty index 0.41 (SD = 0.14) vs. 0.29 (SD = 0.15), respectively; *p* = 0.001] (Table 1). No relevant differences were found regarding chronological age and body weight.

At the 6-month follow-up, patients facing an adverse event exhibited poorer physical performance [median SPPB 5 (IQR = 6) vs. 6.5 (IQR = 8.25), respectively; *p* = 0.04], a higher risk of malnutrition [mean MNA 11.8 (SD = 2.0) vs. 12.6 (SD = 1.5), respectively; *p* = 0.04] and polypharmacotherapy [median number of drugs 8 (IQR = 5) vs. 5 (IQR = 4), respectively; *p* < 0.001] compared to counterparts (Table 3).

Despite not reaching statistical significance, we found a relevant decrease in BADLs in patients experiencing adverse events compared to peers (12.5% vs. 5.2%, *p* = 0.22). Regarding comorbidities, patients with history of chronic heart failure [11 (45.8%) vs. 16 (21.1%), respectively; *p* = 0.01] or chronic renal failure [6 (25%) vs. 6 (7.9%), respectively; *p* = 0.02] had a higher prevalence of adverse events than those without (Table 1). Dyspnea (37%) and fatigue (27%) were the most commonly self-reporte persistent symptoms, followed by impaired walking (15%), impaired memory (11%), reduced power of concentration (9%), cough (9%), loss of taste and smell (9% for both), muscle pain (6%) and arthralgia (6%) (Table 3). Via logistic multivariate analysis, heart failure, chronic renal failure, CIRS and FI emerged as independent predictors of 12-month serious adverse events requiring hospitalization, even after extensive adjustment for confounders [heart failure aOR 3.00 (95% CI 1.10–8.16), *p* = 0.031; chronic renal failure aOR 3.83 (95% CI 1.09–13.43), *p* = 0.036; CIRS aOR 1.95 (95% CI 1.28–2.97), *p* = 0.002; frailty status aOR 7.77 (95% CI 2.13–28.27), *p* = 0.002] (Figure 1). Finally, the inferential plot of the relationship between baseline FI score and risk of adverse events shows that, starting from FI = 0.4, each 10% increase in FI score leads to 13% increased odds of a serious adverse event requiring rehospitalization (Figure 2).

## 4. Discussion

Despite older patients having the highest risk of adverse outcomes due to COVID-19 and facing a high short-term mortality [19], few studies have specifically sought to evaluate post-COVID-19 re-hospitalization trends in the advanced age [6,20]. Hence, there is an unmet clinical need to identify potentially modifiable predictors of short-term adverse events in the elderly.

### 4.1. Prevalence and Causes of 12-Month Adverse Events Requiring Rehospitalization

In our cohort of older patients previously hospitalized for COVID-19, almost one-fourth reported an adverse event that required re-hospitalization. Interestingly, two-thirds of hospital re-admissions occurred within the first three months following discharge for COVID-19, thus suggesting that physicians and healthcare policymakers hold a close-range follow-up after hospital discharge of older COVID-19 patients. Acutely decompensated heart failure, pneumonia and bone fractures were the three most common causes of hospital re-admission, while history of both chronic heart failure and renal disease resulted in the comorbidities most associated with serious adverse events.

### 4.2. Risk Factors of Adverse Events Requiring Re-Hospitalization

Of note, COVID-19 severity was not associated with an increased risk of adverse events after discharge; conversely, the frailty index score emerged as a strong predictor of re-hospitalization. In particular, frail patients exhibited a seven-fold higher prevalence of rehospitalization, reflecting the known association between poor outcome and frailty status in older COVID-19 survivors [21].

In our prospective study, we found a two-times higher prevalence of re-hospitalization as compared with previous reports, showing a ten-percent prevalence rate after discharge for COVID-19 [9]. This finding could be explained by the higher mean age of our cohort, albeit conflicting data are present in the literature regarding the prognostic impact of age on re-hospitalization. However, the study by Carrillo et al. [22] including patients older than 70 years found a 20% incidence of rehospitalization at the 3-month follow-up, thus confirming the role of aging in poor post-COVID outcomes.

Older patients are frequently affected by comorbidities and exposed to poly-pharmacy and, according to previous reports [23,24,25], both conditions are associated with an increased risk of hospital re-admission. In agreement with Bowles et al. [9], chronic renal failure and heart failure were found to be the main clinical features associated with re-hospitalization, both yielding an almost three-fold higher prevalence of rehospitalization as compared to controls. As reported by Rey et al. [26] the potential of SARS-CoV-2 to produce myocardial injury may result in higher mortality and more frequent acute decompensation, particularly in older patients with impaired baseline cardiopulmonary reserves. Accordingly, acute heart failure was found to be the most frequent cause of re-hospitalization after COVID-19, thus confirming acute heart failure as the most common adverse outcome after COVID-19 hospital discharge [9]. Whether this relationship might be explained by direct myocardium damage by COVID-19 or by a reacutization of preexisting not-well-compensated heart failure is still debated [26,27].

On the other hand, regarding the prognostic role of chronic renal failure, SARS-CoV-2 has been demonstrated to infect and replicate directly in renal cells [28], with potential development or worsening of kidney disease [29]. However, the actual long-term implication of kidney involvement by SARS-CoV-2 infection is still unclear. In their review, Touyz et al. [30] highlighted that 25% to 35% of COVID-19 patients did not recover to baseline renal function and develop de novo renal failure or worsened previous chronic kidney disease [31]. Overall, these findings suggest that older people with both chronic heart failure and renal disease should be promptly re-evaluated in order to prevent post-COVID-19 serious adverse events.

To the best of our knowledge, this is the first study demonstrating a significant proportion of femoral fractures following hospital discharge for COVID-19 disease. Of note, our study focused specifically on the oldest old, namely the subset of patients most at risk for accidental falls [32], suggesting the need for a prompt post-discharge rehabilitation care and social aid. In this context, particularly during the first and second COVID-19 waves in Italy, the pandemic brought unprecedent disruption to the provision of mental health care [33], with an increased risk of poor outcomes in the frailest and oldest population, leading to the so-called “social connectivity paradox” [34].

### 4.3. Impact on Functional Performance

Regarding CGA items, patients who experienced adverse events presented a greater disability burden, whereas no relevant difference was found in terms of chronological age between groups. These results are in line with Hewitt et al., reporting that disease outcomes were better predicted by frailty than age itself [21]. As expected, re-hospitalized patients differed significantly in terms of nutritional status and physical performance compared to counterparts at the six-month follow-up, thus highlighting the risk of sarcopenia and mobility impairment following hospital admission.

The most common COVID-19 related symptoms reported in our patients were dyspnea and fatigue followed by impaired walking; interestingly, this latter symptom differed significantly between re-hospitalized patients compared to counterparts. Wostin et al. hypothesized that fatigue and other symptoms often observed in patients with long-COVID syndrome are similar to chronic fatigue syndrome (CFS) [35], a complex disease often underdiagnosed and related to several viruses such as Epstein-Barr, cytomegalovirus, enterovirus and herpes virus [36]. Indeed, the persistence of fatigue in older patients could be explained by the decline in muscle performance caused by virus-induced damage to muscle cells and surrounding vessels [27]. Accordingly, although not statistically significant, we detected higher HGS test values in patients not having a hospital re-admission compared to re-admitted ones.

A few limitations of the current study need to be considered. Firstly, with a small sample size and the lack of a control cohort of patients without COVID-19, caution must be applied, as the finding might be not generalizable to older patients as a whole; secondly, we might have encountered a selection bias during the follow-up, with the most robust patients more likely to present at ambulatory visits compared to frailer peers; thirdly, due to logistical issues raised during the third wave of COVID-19 pandemic, we could not directly assess the one-year mobility function of the enrolled patients. That being said, we did not find any new onset of disability, re-hospitalization or death via phone interviews. Notwithstanding these limitations, our study provides a comprehensive description of demographic data, comorbidities, presenting symptoms and functional outcomes of older COVID-19 survivors.

## 5. Conclusions

In patients previously hospitalized for COVID-19, baseline frailty degree and specific comorbidities such as chronic heart failure or kidney failure emerged as independent predictors of post-discharge serious adverse events requiring hospitalization, thus identifying high-risk subjects who may benefit from a close-range follow-up strategy of care.

## Figures and Tables

**Figure 1 geriatrics-08-00010-f001:**
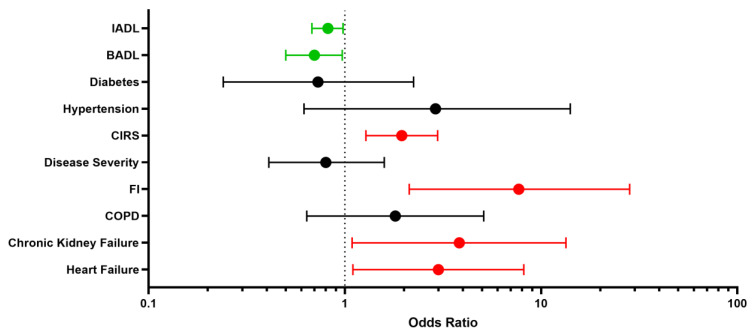
Determinants of re-hospitalization: risk factors (red) and protective factors (green). Forest plot. **BADL** indicates **B**asic **A**ctivities of **D**aily **L**iving; **IADL**, **I**nstrumental **A**ctivities of **D**aily **L**iving; **CIRS**, **C**umulative **I**llness **R**ating **S**cale; **FI**, **F**railty **I**ndex; **COPD**, **C**hronic **O**bstructive **P**ulmonary **D**isease.

**Figure 2 geriatrics-08-00010-f002:**
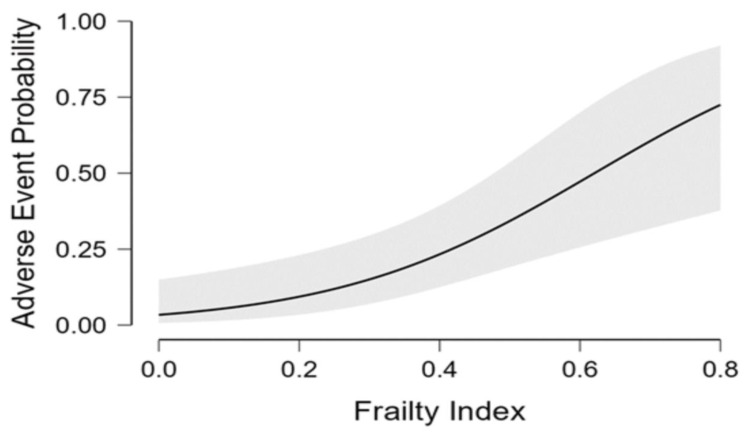
Regression analysis inferential plot. Relationship between baseline frailty index score and odds of serious adverse event requiring re-hospitalization.

**Table 1 geriatrics-08-00010-t001:** Comprehensive Geriatric Assessment (CGA) at baseline.

Comprehensive Geriatric Assessment (CGA)	Whole Cohort (n = 100)	Patients with Adverse Events (n = 24)	Controls (n = 76)	*p*
**Gender F (%)**	42 (42)	10 (41.7)	32 (42.1)	0.97
**Age** **[mean years (SD)]**	85 (4)	85.6 (4.1)	84.8 (4)	0.40
**BMI** **[mean (SD)]**	25.8 (3.9)	25.8 (4.5)	26.2 (3.8)	0.67
**BADL** **[median (IQR)]**	5 (1)	5 (3.25)	6 (1)	0.02
**IADL** **[median (IQR)]**	6.5 (5)	4.5 (6)	7 (4)	0.03
**CIRS** **[median (IQR)]**	3 (2)	4 (1.25)	2 (2)	0.002
**Frailty index** **[mean (SD)]**	0.32 (0.16)	0.41 (0.14)	0.29 (0.15)	0.001
**Not frail (%)** **Frail (%)**	44 (44) 56 (56)	3 (12.5) 21 (87.5)	41 (54) 35 (46)	0.001
**Hypertension (%)**	81 (81)	22 (91.7)	59 (77.6)	0.12
**Heart failure (%)**	27 (27)	11 (45.8)	16 (21.1)	0.01
**Ischemic heart disease (%)**	21 (21)	6 (25)	15 (19.7)	0.58
**Diabetes (%)**	25 (25)	5 (20.8)	20 (26.3)	0.58
**COPD (%)**	25 (25)	8 (33)	17 (22.3)	0.28
**Stroke (%)**	12 (12)	3 (12.5)	9 (11.8)	0.93
**Chronic renal failure (%)**	12 (12)	6 (25)	6 (7.9)	**0.02**
**Previous cancer (%)**	30 (30)	8 (33.3)	22 (28.9)	0.68
**COVID-19 severity** **No supplemental oxygen (%)** **Low flow oxygen (%)** **High flow oxygen (%)**	12 (12) 21 (21) 67 (67)	4 (17.4) 4 (17.4) 16 (65.2)	8 (9.4) 17 (22.3) 51 (67)	0.56

Continuous variables are expressed as mean SD or median with IQR properly. **BMI** indicates **B**ody **M**ass **I**ndex; **BADL**, **B**asic **A**ctivities of **D**aily **L**iving; **IADL**, **I**nstrumental **A**ctivities of **D**aily **L**iving; **CIRS**, **C**umulative **I**llness **R**ating **S**cale; **COPD**, **C**hronic **O**bstructive **P**ulmonary **D**isease.

**Table 2 geriatrics-08-00010-t002:** Serious adverse events requiring hospitalization.

Total Adverse Events	26
**Heart failure (%)**	4 (15.3)
**Pneumonia (%)**	4 (15.3)
**Bone fracture (%)**	4 (15.3)
**Acute kidney failure (%)**	3 (11.5)
**Neurological (%)**	3 (11.5)
**Sepsis (%)**	2 (7.6)
**Syncope (%)**	1 (3.8)
**Hematuria (%)**	1 (3.8)
**Pulmonary embolism (%)**	1 (3.8)
**Subdural hematoma (%)**	1 (3.8)
**Atrial Fibrillation (%)**	1 (3.8)
**Pericarditis (%)**	1 (3.8)

**Table 3 geriatrics-08-00010-t003:** Comprehensive Geriatric Assessment and COVID-19 related symptoms at 6-month follow-up.

	Whole Cohort (n = 100)	Patients with Adverse Events (n = 24)	Controls (n = 76)	*p*
**Hand Grip Strength** **[mean Kg (SD)]**	20.7 (7.5)	20 (6.9)	21 (7.8)	0.59
**MNA** **[mean (SD)]**	12.4 (1.7)	11.79 (2.02)	12.59 (1.5)	0.04
**N° of drugs** **[median (IQR)]**	6 (4)	8 (5)	5 (4)	<0.001
**SPPB** **[median (IQR)]**	6 (8)	5 (6)	6.5 (8.25)	0.04
**SPSMQ** **[median (IQR)]**	2 (3)	2 (3.25)	1.5 (2)	0.15
**Loss of BADL (%)**	7 (7)	3 (12.5)	4 (5.2)	0.23
**Dyspnea (%)**	37 (37)	11 (45.8)	26 (34.2)	0.18
**Cough (%)**	9 (9)	3 (12.5)	6 (7.9)	0.78
**Pharyngodynia (%)**	3 (3)	2 (8.3)	1 (1.3)	0.21
**Muscle pain (%)**	6 (6)	3 (12.5)	3 (3.9)	0.12
**Arthralgia (%)**	6 (6)	2 (8.3)	4 (5.2)	0.85
**Fatigue (%)**	27 (27)	9 (37.5)	18 (23.8)	0.35
**Heartburn (%)**	4 (4)	1 (4.1)	3 (3.9)	0.96
**Loss of taste (%)**	9 (9)	1 (4.1)	8 (10.5)	0.61
**Loss of smell (%)**	9 (9)	2 (8.3)	7 (9.2)	0.91
**Deambulation impairment (%)**	15 (15)	8 (33.3)	7 (9.2)	**0.01**
**Concentration impairment (%)**	9 (9)	1 (4.1)	8 (10.5)	0.56
**Memory impairment (%)**	11 (11)	3 (12.5)	8 (10.5)	0.91

Continuous variables are expressed as mean SD or median with IQR properly. **MNA** indicates **M**ini **N**utritional **A**ssessment; **SPPB**, **S**hort **P**hysical **P**erformance **B**attery; **SPSMQ**, **S**hort **P**ortable **M**ental **S**tatus **Q**uestionnaire; **BADL**, **B**asic **A**ctivities of **D**aily **L**iving.

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
