# Peer review of "Determinants of 1-Year Adverse Event Requiring Re-Hospitalization in COVID-19 Oldest Old Survivors"

_geriatrics, 2023, doi:10.3390/geriatrics8010010_

Round 1

Reviewer 1 Report

Dear Authors, 

I had the pleasure to review your manuscript. I can only congratulate you and hope that your article will be published fast.

Best Regards

Author Response

We would like to thank the Reviewer for the comment; we are grateful and happy for the support

Reviewer 2 Report

This study explored the Determinants of 1-Year Adverse Event Requiring Re-hospitalization in COVID-19 Oldest Old Survivors, through a 12-month computerized archive and phone interviews. This study helps to determine which elderly people need to pay more attention after being infected with COVID-19. However, there are some issues need to be addressed before publication.

1.     My main concern is the lack of a control group in this article. The research object of this paper is 100 patients discharged after hospitalization for COVID-19. I think this study needs a group of subjects with the same conditions but not infected with COVID-19. Otherwise, how to determine that the follow-up analysis results are all caused by COVID-19 infection?

2.     Clear figures are necessary, for example, Figure 1 is ambiguous

3.     Two or three aims are listed in the end of Section 1. Perhaps it would be better to list the research results of these several aims in the later part of the article.

Author Response

My main concern is the lack of a control group in this article. The research object of this paper is 100 patients discharged after hospitalization for COVID-19. I think this study needs a group of subjects with the same conditions but not infected with COVID-19. Otherwise, how to determine that the follow-up analysis results are all caused by COVID-19 infection?

Response: We are thankful to the Reviewer for pointing out this important issue. We agree that the lack of a control cohort age and gender matched from the same time-window could be seen as a limitation of our study. However, given the dramatic impact of COVID-19 on acutely ill older adults, the aim was to evaluate, through follow-up, the recovery of this particular cluster of patients in light of specific predictors of adverse events requiring hospitalization useful for clinical practice. Therefore, the main focus was the phenotype of the patients with COVID-19, and the possible impact of it on the outcome and recovery. In our opinion, we find interesting and relevant the finding that a higher risk of re-hospitalization was not so much attributable to the Covid-19 severity as to the co-existing presence of particular chronic diseases and frailty degree, suggesting that, in the medium-long term, SARS-CoV-2 infection could exert its effects indirectly by worsening pre-existing patient’s impairment. Moreover, when the study was conceived, it was impossible to enroll patients as controls due to the extraordinary measures put in place for the COVID-19 pandemic, which led to the conversion of our geriatrics ward into a COVID-19 one. We mentioned this limitation in the dedicated part of the discussion (Line 266 of the tracked version of the manuscript).

Clear figures are necessary, for example, Figure 1 is ambiguous

Response: We thank the Reviewer for the suggestion. To make figure 1 clearer, we have updated the caption as follows:

Figure 1. Determinants of re-hospitalization: risk factors (red) and protective factors (green). Forest plot.

Two or three aims are listed in the end of Section 1. Perhaps it would be better to list the research results of these several aims in the later part of the article.

Response: thank you for the suggestion. As kindly suggested, we have listed the research results in the discussion.